# Optimal control of a quantum sensor: A fast algorithm based on an analytic solution

Santiago Hernández-Gómez[1,2,3], Federico Balducci[4,5,6], Giovanni Fasiolo[7], Paola Cappellaro[8], Nicole Fabbri[2,9]* and Antonello Scardicchio[4,5]†

**1** Research Laboratory of Electronics, Massachusetts Institute of Technology, Cambridge, MA 02139
**2** European Laboratory for Non-linear Spectroscopy (LENS), Università di Firenze, I-50019 Sesto Fiorentino, Italy
**3** Dipartimento di Fisica e Astronomia, Università di Firenze, I-50019, Sesto Fiorentino, Italy
**4** The Abdus Salam ICTP – Strada Costiera 11, 34151, Trieste, Italy
**5** INFN Sezione di Trieste – Via Valerio 2, 34127 Trieste, Italy
**6** SISSA – via Bonomea 265, 34136, Trieste, Italy
**7** Università degli studi di Trieste, Piazzale Europa 1, 34127, Trieste, Italy
**8** Department of Nuclear Science and Engineering, Department of Physics, Massachusetts Institute of Technology, Cambridge, MA 02139
**9** Istituto Nazionale di Ottica del Consiglio Nazionale delle Ricerche (CNR-INO), I-50019 Sesto Fiorentino, Italy

* fabbri@lens.unifi.it , † ascardic@ictp.it

## Abstract

**Quantum sensors can show unprecedented sensitivities, provided they are controlled in a very specific, optimal way. Here, we consider a spin sensor of time-varying fields in the presence of dephasing noise, and we show that the problem of finding the optimal pulsed control field can be mapped to the determination of the ground state of a spin chain. We find an approximate but analytic solution of this problem, which provides a *lower bound* for the sensor sensitivity, and a pulsed control very close to optimal, which we further use as initial guess for realizing a fast simulated annealing algorithm. We experimentally demonstrate the sensitivity improvement for a spin-qubit magnetometer based on a nitrogen-vacancy center in diamond.**

# 1   Introduction

Quantum systems are notoriously sensitive to external influences. This sensitivity is a core element in the development of quantum technologies, as is the case of quantum sensing, which takes advantage of quantum coherence to detect weak or nanoscale signals. Quantum sensing devices can in principle attain precision, accuracy, and repeatability reaching fundamental limits [1,2]. However, the extreme sensitivity to external perturbations also causes the quantum sensor to couple with detrimental noise sources that induce decoherence, therefore limiting the interaction time with the target signal.

Here, we introduce a method to find optimal control protocols [3] for ac quantum sensing in the presence of dephasing noise. Such optimization problem is in general a complex classical problem. Our method, that draws an analogy between pulsed dynamical decoupling (DD) protocols [4–8] and spin glass systems [9], maximizes the phase acquired by the quantum sensor due to the target ac field while minimizing the noise detrimental effect. The optimal control fields yield an improved sensitivity with respect to commonly used protocols, as we experimentally demonstrate using a spin-qubit magnetometer based on a Nitrogen-Vacancy (NV) center in diamond [10–14].

More in detail, we find that the problem of optimizing the control protocol for our quantum sensor is homologous to that of finding the ground state of a classical Ising spin Hamiltonian, as depicted in Fig. 1. The control $\pi$-pulse times correspond to the locations in the chain of the domain walls. The couplings between the model spins, which encode the noise autocorrelation, are of both signs, and this is customary in optimization problems. The antiferromagnetic couplings capture the frustration between the different terms in the Hamiltonian, which then *prima facie* is that of a spin-glass model—which does not mean that there is a spin-glass *phase* at low temperature (see later).

The study of optimization problems in statistical physics is a large field of research in disordered systems, with far-reaching connections to the physics of spin glasses [15,16] and other frustrated, classical and quantum models [17–23]. Optimization problems in quantum control can show some degree of frustration, with terms that compete in a similar way in which ferromagnetic and anti-ferromagnetic bonds compete in spin glasses [24]. We find, however, that in the specific case of the optimal control of a qubit sensor, by trading the Ising $\mathbb{Z}_2$ spins for the continuous spins of a spherical model (SM) [25,26] one gets rid of frustration

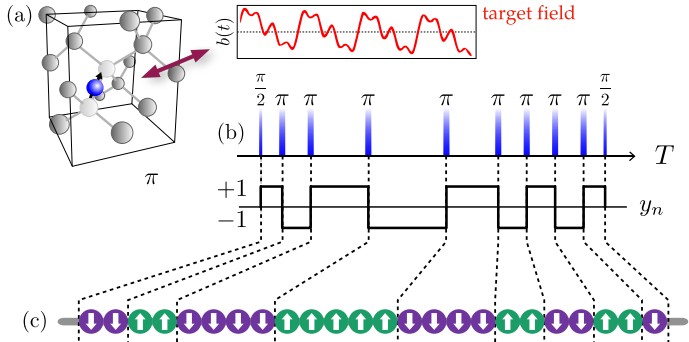

Figure 1: (a) A single spin sensor is used to detect an AC target magnetic field $b(t)$. (b) An optimal control field applied to the spin sensor increases its coherence, hence improving its sensitivity. (c) The difficult problem of finding an optimal control sequence can be mapped into a problem of finding the ground state of a virtual spin chain.

altogether, and the model shows little signs of competing equilibria at low temperature, typical of replica-symmetry-broken phases [9, 27]. Since the ground state of the spherical model can be found analytically if the spectra of the signal and of the noise are known, we obtain from this both a *lower bound* for the sensitivity [1], and a quasi-optimal controlled pulsed field. This quasi-optimal sequence can then be fed to a simulated annealing (SA) algorithm [28–31], in order to find the optimal one with little computational effort. Such annealed sequences show, in agreement with the experiments, very good sensitivities (only about **20%** worse than the bound). Our method is thus superior to standard DD protocols as Carr-Purcell (CP), or minimal generalizations of the latter to colored signals, as we discuss in detail. Finally, to show the unparalleled performance of the algorithm, which can open the door to real-time optimization in sensing, we run it on a Raspberry Pi microcomputer, where it takes milliseconds to find the optimal solutions.

## 2   Optimized dynamical decoupling for sensing

We consider a single spin-qubit sensor of time-varying magnetic fields, in the presence of dephasing noise. This quantum sensing task can be described as a compromise between spin phase accumulation due to the external target field to be measured $b(t) \equiv bh(t)$, and refocusing of the non-Markovian noise, obtained via dynamical decoupling (DD) protocols [4–8]. Above, $b$ is the magnetic field strength to be detected, and $h(t)$ a known, dimensionless function specifying its time dependence.

As in Hahn's echo [32–34], a DD sequence is implemented by applying sets of $n$ $\pi$-pulses that act as time reversal for the phase acquired by the qubit during its free evolution, and can be described by a modulation function $y : [0, T] \ni t \mapsto \{-1, 1\}$ (see Fig. 1b). The DD sequence is embedded within a Ramsey interferometer, hence the qubit coherence is mapped onto the probability of the qubit to populate the excited state $|1\rangle$:

$$P(T, b) = \text{Tr}[\rho |1\rangle\langle 1|] = \frac{1}{2}\left(1 + e^{-\chi(T)} \cos \varphi(T, b)\right). \tag{1}$$

---

[1]The lower bound is *not* related to the Cramér-Rao bound, since the latter is used to define the sensitivity itself [see Supplementary Information].

Here, $\varphi$ is the phase acquired by the qubit during the sensing time $T$:

$$\varphi(T, b) = b\gamma \int_0^T dt\, h(t)\, y(t), \tag{2}$$

with $\gamma$ the coupling to the field (e.g., the electronic gyromagnetic ratio of the spin sensor). The noise-induced decoherence function

$$\chi(T) \equiv \frac{1}{\pi} \int \frac{d\omega}{\omega^2} S(\omega) |Y(T, \omega)|^2 \tag{3}$$

is the convolution between the noise spectral density (NSD) $S(\omega)$ and the filter function $Y(T, \omega) = i\omega \int_0^T dt\, e^{-i\omega t} y(t)$. Note that we neglect the effect of the target field on the noise source [35] and we assume the noise to be a stationary Gaussian process.

Dynamical decoupling is a very versatile control technique, with a virtually infinite space of degrees of freedom spanned by all the possible distributions of $\pi$ pulses, even at finite sensing time $T$. One of the most common DD sequences is the Carr-Purcell (CP) sequence [33, 34], formed by a set of equidistant pulses. Non-equidistant sequences have been proposed and experimentally tested, e.g. in Refs. [7, 36–40]. Each of these sequences has internal degrees of freedom, that can be tuned to increase the sensing capabilities for specific target fields. Another example is what we call the "generalized Carr-Purcell" (gCP) protocol, in which $\pi$ pulses are applied when the signal $b(t)$ changes sign, i.e. in correspondence to its zeros. All these DD sequences are already optimal for specific target fields that do not overlap significantly with the noise. However, as the complexity of the target field increases, it increases also the difficulty to find a pulse sequence that successfully filters out the noise components, while still maintaining the sensitivity to the target field.

A possible approach is to use an optimization algorithm, to find a $\pi$-pulse sequence that optimizes a desired figure of merit, for example the sensitivity, i.e. the smallest detectable signal. This concept was proposed and demonstrated experimentally for an NV center used as a quantum magnetometer [41]. Despite the achieved improvements, the computational complexity of the above optimization problem limited its applicability. Indeed, the optimization cost function, the sensitivity $\eta$, defined as [1, 42]

$$\eta = \frac{e^{\chi(T)}}{|\varphi(T)/b|} \sqrt{T}, \tag{4}$$

(see also the Supplementary Information for a derivation) is a compromise between noise cancellation and target ac field encoding, and it is hard to optimize.

In our approach, instead, we recast the cost function $\eta$ as the Hamiltonian of a classical Ising spin system. In this way, the continuous optimization problem for the minimization of the sensitivity of a NV-center magnetometer is re-interpreted as a discrete energy minimization problem. Specifically, we define the new cost function to be the (dimensionless) logarithmic sensitivity

$$\epsilon = \log\left(\eta\gamma\sqrt{T}\right) = \chi(T) - \log\left|\frac{\varphi(T)}{T\gamma b}\right|, \tag{5}$$

and we show in Sec. 3 that upon time discretization $\epsilon$ becomes an Ising Hamiltonian, albeit with sign-alternating, long-range interactions and a peculiar logarithmic field-spin coupling. Before doing that, however, we show how the problem can be tackled in continuous time, and by means of a reasonable approximation.

### 3 A variational approach

Our task is to find the optimal function $y(t)$ which minimizes the sensitivity $\eta$, Eq. (4), or the logarithmic sensitivity $\epsilon$, Eq. (5). First of all, we anticipate why simple choices for $y(t)$ do not yield good results for generic sensing tasks. Looking at Eqs. (2)–(4), one understands that the minimum detectable signal $\eta$ is determined by a competition of the signal, through $\varphi$, and the noise, through $\chi$. Commonly used DD protocols, as CP sequences, focus only on the properties of the signal, trying to amplify it irrespective of the noise (or assuming the zero-to-low-frequency noise). So, either using a CP sequence to amplify one frequency the signal is composed of, or taking $y(t) \propto h(t)$ to mimic as close possible the signal (the strategy dubbed gCP above), fail when the noise and the signal share common frequencies. Nevertheless, with the procedure outlined below, we show how it is possible to "orthogonalize" the DD sequence wrt. the noise to minimize the overlap $\chi$, while keeping it "parallel" to the signal to maximize $\varphi$. In passing, we obtain useful analytical results that allow us to assess the performance of our method.

Let us rewrite $\epsilon$ as [see Eqs. (2),(3) and (5)]

$$\epsilon[y] = \frac{1}{2} \int_{[0,T]^2} dt\, dt'\, y(t) J(t,t') y(t') - \log \left| \frac{1}{T} \int_0^T dt\, h(t) y(t) \right|, \qquad (6)$$

with

$$J(t,t') = \frac{2}{\pi} \int d\omega\, \cos(\omega(t'-t)) S(\omega). \qquad (7)$$

$J(t,t')$ is the noise autocorrelation function, which depends only on the difference $t - t'$ by stationarity of the noise. Notice also that $J$ is a positive operator even though $J(t',t)$ can take up any values in $\mathbb{R}$. Then, in order to find $y(t)$ that minimizes $\epsilon$, we start by imposing the constraint $y(t)^2 = 1$ for all $t$ via a continuous set of Lagrange multipliers, i.e. via a function $\lambda(t)$:

$$F[y,\lambda] = \epsilon[y] + \frac{1}{2} \int_0^T dt\, \lambda(t)\left(y(t)^2 - 1\right). \qquad (8)$$

We need to find the stationary point of $F[y]$ w.r.t. $y(t)$ and $\lambda(t)$. Formally, the saddle point equations are

$$\frac{\delta F}{\delta y(t)} = \int_0^T dt'\left[J(t,t') + \lambda(t)\delta(t-t')\right] y(t') - \frac{h(t)}{\int_0^T dt\, h(t') y(t')} = 0, \qquad (9)$$

$$\frac{\delta F}{\delta \lambda(t)} = y^2(t) - 1 = 0. \qquad (10)$$

One can see that the extreme w.r.t. $\lambda$ simply gives the constraint. The formal solution of the above equations is

$$y(t) = \frac{1}{D} \int_0^T dt'\left(\frac{1}{J+\lambda}\right)_{t,t'} h(t'), \qquad (11)$$

where $\lambda$ stands for the diagonal operator $\lambda(t)\delta(t-t')$, and

$$D = \int_0^T dt\, h(t) y(t) = \frac{1}{D} \int_0^T dt\, dt'\, h(t)\left(\frac{1}{J+\lambda}\right)_{t,t'} h(t'), \qquad (12)$$

$$\Longrightarrow D = \left(\int_0^T dt\, dt'\, h(t)\left(\frac{1}{J+\lambda}\right)_{t,t'} h(t')\right)^{1/2}. \qquad (13)$$

The quantity $D$ can be interpreted as a self-consistent normalization for $y(t)$. By plugging Eq. (11) in Eq. (8), one can express the cost function at the saddle as

$$
\begin{aligned}
F &= \frac{1}{2} \int_{[0,T]^2} dt\, dt'\, y(t) J(t,t') y(t') - \log\left|\frac{1}{T}\int_0^T dt\, h(t) y(t)\right| + \frac{1}{2}\int_0^T dt\, \lambda(t)\left(y(t)^2 - 1\right) \\
&= \frac{1}{2D^2}\int_{[0,T]^2} dt\, dt'\, h(t)\left(\frac{1}{J+\lambda}\right)_{t,t'} h(t') - \log\left|\frac{1}{DT}\int_0^T dt\, dt'\, h(t)\left(\frac{1}{J+\lambda}\right)_{t,t'} h(t')\right| \\
&\quad -\frac{1}{2}\int_0^T dt\, \lambda(t) \\
&= \frac{1}{2} - \log\left|\frac{D}{T}\right| - \frac{1}{2}\int_0^T dt\, \lambda(t).
\end{aligned}
\tag{14}
$$

The last expression is a function of $\lambda(t)$ only and one can, in principle, find the saddle point of it and substitute it in Eq. (11) to obtain the optimum DD sequence.

Short of solving exactly the model in Eq. (8), we can get good results to guide the experiment by simplifying the space in which we are searching for the minimum. We can do this in two ways: either we keep $y(t)$ defined on $\mathbb{R}$ (i.e. we keep the time continuum) and we give more structure to $\lambda(t)$, or we discretize time and enforce the constraint $y(t)^2 = 1$ exactly (therefore getting rid of $\lambda$). These two approaches will be implemented in the following.

## 3.1 Spherical approximation

In order to make progress, we substitute for the moment the constraint $y(t)^2 = 1$, for all $t$, with the constraint

$$
\frac{1}{T}\int_0^T dt\, y^2(t) = 1.
\tag{15}
$$

This is equivalent to finding the stationary point of $F[y,\lambda]$, Eq. (8), in the subspace in which $\lambda(t) \equiv \lambda$. We call the resulting approximation *spherical model* (SM) [2], taking inspiration from the physics of spin glasses [25,26].

Spherical models are often good mean field models of spin glasses and of their dynamics [25,26,43], and this case will prove to be of similar nature despite the unusual logarithmic field coupling term. By setting $\lambda(t) \to \lambda$ we have the function of the single parameter

$$
\epsilon_{\text{SM}}(\lambda) = \frac{1}{2} - \frac{T}{2}\lambda - \frac{1}{2}\log\left|\frac{1}{T^2}\int_0^T dt\, dt'\, h(t)\left(\frac{1}{J+\lambda}\right)_{t,t'} h(t')\right|
\tag{16}
$$

where $J + \lambda$ is the operator with integral kernel $J(t',t) + \lambda\delta(t'-t)$, as above. Minimizing w.r.t. $\lambda \in \mathbb{R}$, one finds a theoretical lower bound on the sensitivity: $\eta > \eta_{\text{SM}} = e^{\epsilon_{\text{SM}}}/\gamma\sqrt{T}$. This is a lower bound for the sensitivity because the minimum of $\epsilon_{\text{SM}}$ corresponds to a $y(t)$ over a *larger* space of functions (Eq. (15) is weaker than constraint $y(t)^2 = 1$), as shown schematically in Fig. 2. In principle the bound is not sharp, however it provides a quick and accurate measure of the goodness of our results. Moreover, we have found by experience that it is *in practice* pretty close to being sharp and that it can hardly be improved analytically by adding more freedom to the function $\lambda(t)$ beyond the constant $\lambda(t) = \lambda$. For example the test function $\lambda(t) = \lambda_1\chi_{[0,T/2]}(t) + \lambda_2\chi_{[T/2,T]}(t)$ ($\chi_{[a,b]}$ is the characteristic function of the interval $[a,b]$), giving a two-parameters space $(\lambda_1,\lambda_2)$ for minimization, gives at most a few percent increase on the bound on $\eta$. We therefore use it *as if it were sharp*.

---

[2]The name "spherical" comes from the fact that, after having discretized time in $N$ different, equally spaced values $t_i = i\Delta t$, the constraint in Eq. (15) puts the variable $y(t)$ on a $N$-dimensional sphere, where $N = T/\Delta t$.

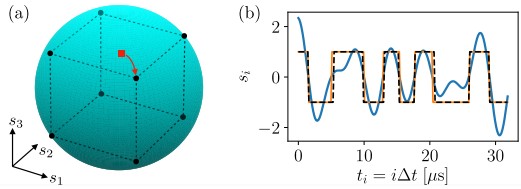

Figure 2: (a) Sketch of the spherical model. The $N$-dimensional, hyper-spherical surface (in blue) strictly contains the hypercube $\{+1,-1\}^N$ (black dots), each point of which encodes the configuration of classical spins in the Ising model. Therefore, the solution of the spherical model (red square) is, in general, not a point on the hypercube, but it can be projected (arrow) onto the latter, giving a good value for the sensitivity. (b) Comparison between solutions for **200** spins ($T = 32\,\mu s$, $\Delta t = 0.16\,\mu s$). The continuous spins $s_i \in \mathbb{R}$ (blue line) can be converted into Ising spins $s_i = \pm 1$, necessary for the $\pi$-pulses, by using the sign function (orange line): this step corresponds to the projection (arrow) in panel (a). The sensitivity can be improved further with a few iterations of SA to get a close-by sequence (dashed black line). In this example, the trichromatic target signal and the noise are equal to the ones used in the experiment (see text).

One can define for any DD sequence the dimensionless quantity $\eta_{SM}/\eta < 1$. We will see in the next section how different approximate solutions give different values of this quantity. Moreover, we will see how the solution of the SM, although not *per se* a DD sequence, can function as a starting point for finding an optimal DD sequence.

## 3.2  Time discretization and Simulated Annealing

Let us focus now on the second method: time discretization. We discretize the sensing time $T$ into small time intervals $\Delta t$, to obtain a sequence of times $t_i = i\Delta t$ with $i \in 1, ..., N = T/\Delta t$. The interval $\Delta t$ is the smallest time we allow the $\pi$-pulses of the DD sequence to be separated by. Apart from the physical limit given by the experimental apparatus, which sets a minimal $\Delta t$, one does not expect to need in the optimal solution $\pi$-pulses separated by much less than the minimum period of $h(t)$, if it exists (the spectrum of $h(t)$ can extend up to infinite frequency). The modulation function at each of these times is $y(t_i) = \pm 1$, which dictates the sign of the phase acquired by the spin qubit during the time interval $[t_i - \Delta t, t_i]$. We can therefore write the modulation function as

$$y(t) = \sum_{i=1}^{N=T/\Delta t} s_i \chi_{[(i-1)\Delta t, i\Delta t]}(t), \tag{17}$$

where $s_i = \pm 1$, and as before $\chi_{[a,b]}$ is the characteristic function of an interval $[a, b]$. Writing the modulation function in this way allows us to recast Eqs. (2) and (3) respectively as

$$\varphi(T) = T\gamma b \sum_{i=1}^{N} h_i s_i, \tag{18}$$

$$\chi(T) = \frac{1}{2} \sum_{i,j=1}^{N} J_{ij} s_i s_j \tag{19}$$

where

$$h_i = \frac{1}{T} \int_{(i-1)\Delta t}^{i\Delta t} dt\, h(t) \tag{20}$$

represents the interaction with a normalized target ac field, and

$$J_{ij} \equiv \frac{4}{\pi} \int d\omega \frac{[1 - \cos(\omega \Delta t)]}{\omega^2} \cos(\omega(j-i)\Delta t) S(\omega) \tag{21}$$

represents the interaction with the detrimental noise. We can now express the new cost function as

$$\epsilon = \frac{1}{2} \sum_{i,j=1}^{N} J_{ij} s_i s_j - \log \left| \sum_{i=1}^{N} h_i s_i \right| : \tag{22}$$

this closely resembles the Hamiltonian of the Ising spin glass problem for a set of $N$ spins $s_i$. The ground state for this Hamiltonian can be used to obtain a modulation function, therefore a DD sequence, that minimizes the sensitivity $\eta$.

At first sight, minimizing $\epsilon$ in Eq. (22) on the hypercube $\{s_i\} \in \{-1, 1\}^N$ seems a difficult problem, since the couplings $J_{ij}$ can be of both signs. Therefore, one is tempted to use a simulated annealing (SA) minimization algorithm [28–30] to find the minimum of the energy $\epsilon$. However, the performance of SA is strongly affected by the starting configuration both in the final value and, at least as importantly, in the time to reach it. With this in mind we turn to the SM solved in the previous section but with our discretized time, in terms of which the spherical constraint reads $\sum_{i=1}^{N} y_i^2 = N$. In the discretized form, the solution of the SM is (see Eq. (11))

$$y_j = \frac{1}{D} \sum_{k=1}^{N} \frac{e^{i \frac{2\pi j}{N} k}}{\sqrt{N}} \frac{\tilde{h}_k}{\tilde{J}_k + \lambda}. \tag{23}$$

Above, we introduced the Fourier transform of the signal term $\tilde{h}_k = \frac{1}{\sqrt{N}} \sum_j e^{-i \frac{2\pi k}{N} j} h_j$, and of the noise term $\tilde{J}_k = \frac{1}{\sqrt{N}} \sum_j e^{-i \frac{2\pi k}{N} j} J_{i,i-j}$: indeed, since $\lambda(t)$ is constant and $J_{ij}$ depends only on the difference $i - j$, the matrix $J + \lambda$ is diagonal in Fourier space [3]. The value of $\lambda$ is chosen to enforce the spherical constraint, and $D = \left( \sum_{k'} |\tilde{h}_{k'}|^2 / (\tilde{J}_{k'} + \lambda) \right)^{1/2}$, see Eq. (12). One can notice that in Fourier space the optimal solution is aligned with the field, and orthogonal to the noise.

An example solution is shown in Fig. 2. The values of $y_i$ do not form a sequence of $\pm 1$, but the solution is reasonably close to the minimum of the original functional Eq. (22) over the hypercube $\{-1, 1\}^N$. We can now use the solution in Eq. (23) as a starting point to find the optimal sequence $s_i \in \{-1, 1\}$. To do so, we first define $s_i = \text{sign}(y_i) \in \{-1, 1\}$ and then run few steps of SA *moving only the domain walls*, i.e. flipping only spins which are on a sign change: $s_i = -s_{i+1}$. The $\pi$-pulse sequence is, as before, the sequence of times where the spins change sign (the position of the domain walls in the spin chain).

We test our procedure on an ensemble of test cases constructed as follows. The signal is a superposition of monochromatic waves $h(t) = \sum_{n=1}^{N_{\text{freq}}} A_n \cos(\omega_n t + \phi_n)$: we fix $N_{\text{freq}} = 7$ and extract uniformly random frequencies in the interval $[0, 1]$ MHz, uniformly random phases $\phi_n$, and uniformly random amplitudes $A_n$ s.t. $\sum_{n=1}^{N_{\text{freq}}} A_n = 1$. The noise spectrum is instead a gaussian centered at **0.4316** MHz, and with standard deviation **0.016** MHz: thus, it is close to (but a little bit stronger w.r.t.) the experimentally relevant situation discussed in the next session.

First, we use the generalized Carr-Purcell (gCP) protocol introduced above. This procedure is simple but not very effective: on average, it returns between **2/3** and **1/3** of the maximum inverse sensitivity, monotonically decreasing with the time of the sampling (see Fig. 3a). The

---

[3]Strictly speaking, the noise term is represented by a Toeplitz matrix $J_{ij}$, which is diagonalized by the discrete Fourier transform only in the limit $N \to \infty$. However, already at finite $N$ plane waves constitute a reasonable approximation for the eigenvectors [44]. For numerical purposes, any diagonalization routine will suffice.

decay is caused by the fact that the gCP sequences do not take into account the dephasing noise. Hence, as time increases the accumulation of noise by the sensor weakens its sensitivity. Second, we use the solution of the SM, viz. $s_i = \text{sign}(y_i)$, as DD sequence: this gives a better solution, due to the fact that the sequence attempts to partially filter out the noise, but it is still not optimal. The best results, however, are obtained by running a fixed number of steps of SA starting from either a random DD sequence (SA, more on this below), from the gCP DD sequence (gCP+SA), or from the sign(SM) DD sequence (sign(SM)+SA). All these three cases perform the best because the SA algorithm is able to find a good local minima of the optimization landscape. As it is seen in Fig. 3a, the sign(SM)+SA sequence gives the overall best result, with a solution close to the upper bound given by the SM itself (before projecting on the hypercube). It is important to stress that, although the ratio $\eta_{SM}/\eta$ for sign(SM)+SA is close to be constant as a function of time, eventually the sensor will not be able to detect any signals due to decoherence beyond dephasing (not considered in our model), e.g. $T$ is limited by the spin-lattice relaxation time $T_1 \simeq 1$ ms for NV spin sensors at room temperature.

For what concerns the decay of sensitivity for some control fields, our understanding is the following: The gCP case performs the worst because it knows nothing about the noise, and as time increases the gCP solution results in accumulation of noise by the sensor. The situation is better for the sign(SM) case, that encompasses some effect of the noise. The SA cases perform the best because, of course, they represent good local minima of the optimization landscape, that are found by the numerical sampling procedure.

Finally, let us give more details regarding the unbiased SA optimization, called SA above, that starts at infinite temperature from a uniformly random sequence of $s_i = \pm 1$. In this case, to reduce the number of $\pi$ pulses it is necessary to introduce by hand a ferromagnetic coupling term in the Hamiltonian:

$$\epsilon \to \epsilon - K \sum_{i=1}^{N-1} s_i s_{i+1},$$ (24)

with $K > 0$ to be tuned. One can see in Fig. 3b that the best sensitivity is however still obtained with the combination of the SM solution and SA optimization. Additionally, from Fig. 3b one can also understand that the optimal solution represents the best trade-off between number of $\pi$ pulses (which the experimenter would like to maintain low) and sensitivity.

To conclude, we stress that our optimization procedure is very fast, if compared to standard, general-purpose routines. In particular, we were able to run our codes on a Raspberry Pi microcomputer, where the single instance takes $\sim 0.5$ s for the unbiased SA algorithm, and $\sim 0.02$ s for the solution of the SM and subsequent annealing (using $N = 500$ spins). Taking in consideration that few instances of the sign(SM)+SA protocol are sufficient to obtain a good result, while the optimization over the parameter $K$ requires hundreds, if not thousands, of separate SA runs, the gain provided by our method becomes apparent. This fact also opens the door to the miniaturization of the control electronics, in view of possible technological applications of quantum sensing.

# 4 Experiment

While our method is general and applicable to any spin-qubit sensor, we exemplify it through experiments with a single NV center in bulk diamond with naturally abundant $^{13}$C nuclear spins, at room temperature. The ground state electron spin of the NV center can be initialized and measured by exploiting spin-dependent fluorescence, and can be coherently manipulated by microwaves [14]. We consider the two ground-state spin levels, $m_S = 0$ and $m_S = +1$, to form the computational basis of the qubit sensor $\{|0\rangle, |1\rangle\}$ (see Supplementary Information). The main source of noise for the NV spin qubit derives from the collective effect of $^{13}$C

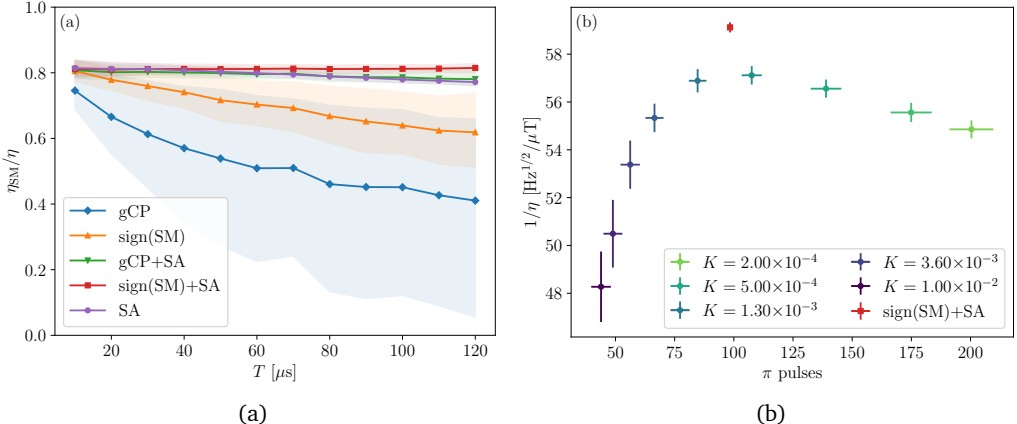

(a)           (b)

Figure 3: (a) Comparison of performances, over a broad ensemble of parameters, for the DD sequences discussed in the main text: generalized Carr-Purcell (gCP), spherical model projected with the **sign** function onto the hypercube (sign(SM)), simulated annealing (SA), and SA optimization starting from gCP and sign(SM). One can see that the best results are obtained for the SA optimization guided by the SM solution. The data refer to the ensemble of random test signals described in the main text: the dots are the average values, and the shaded area represents the 20–80 percentile of the distribution of results. The discretization interval is $\Delta t = 0.1$ $\mu$s. (b) Single instance of a random signal, corresponding to $T = 100$ $\mu$s. We show the average sensitivity and the number of $\pi$ pulses (the error bars correspond to one standard deviation over the ensemble of annealing realizations) of the solutions coming from the unbiased SA, i.e. starting from infinite temperature (purple to green circles), and from the SA guided by the SM solution (red square). The unbiased SA needs a ferromagnetic term $\propto K$, see Eq. (24), with $K$ to be optimized over, in order to keep under control the number of $\pi$ pulses. From this plot, one learns that first, the optimal solution represents also the best trade-off between number of $\pi$ pulses and sensitivity, and second, that the SA optimization guided from the SM performs better, and with less fluctuations. Here, each unbiased SA procedure uses $10^5$ Monte Carlo steps and a power-law temperature ramp, while only $10^3$ steps are needed for the SA from the SM solution.

263  impurities randomly oriented in the diamond lattice.

264  In the presence of a relatively high bias field ($\gtrsim 150$ G), the collective effect of the nuclear
265  spin bath on the NV spin is effectively described as a classical stochastic field, with gaus-
266  sian noise spectral density (NSD) centered at the $^{13}$C Larmor frequency $\nu_L$ [45, 46]. We pre-
267  liminarily characterize the NSD of the NV spin sensor as in Ref. [46]. The direct coupling
268  between the target field and the nuclear spins is negligible due to the small nuclear mag-
269  netic moment [35], and the indirect coupling via the NV electronic spin is also negligible
270  due to the presence of the strong bias field [46]. Therefore, the NV spin dynamics is well
271  described by Eq. (1). For the experiments we present throughout this article we used a bias
272  magnetic field of $403.2(2)$ G, for which the NSD is $S(\omega) = S_0 + A \exp(-(\omega - \omega_L)^2/(2\sigma^2))$, with
273  $S_0 = 0.00119(9)$ MHz, $\omega_L/2\pi \equiv \nu_L = 0.4316(2)$ MHz, $A = 0.52(4)$ MHz, and $\sigma/2\pi = 0.0042(2)$.

274  As a test case for our optimal control method versus standard control, we consider a three-
275  chromatic target signal, with $h(t) = \sum_{i=-1}^{+1} A_i \cos(2\pi\nu_i t)$, where $\nu_i = \{0.1150, 0.2125, 0.1450\}$ MHz
276  are the frequency components, and $A_i = \{0.288, 0.335, 0.377\}$ are the relative amplitudes,
277  respectively for $i = -1, 0, +1$.

278  In Fig. 4(a) we show the NV spin coherence $P(n\tau, b)$ under Carr-Purcell (CP)-type DD
279  control, formed by $n$ pulses with uniform interpulse spacing $\tau = T/n$, as a function of $\tau$. The
280  value of $b$ at the position of the NV defect inside the diamond is obtained from minimizing the
281  squared residuals between experiment (gray bullets) and simulation (gray line), for which $b$
282  is the only free parameter (see Supplementary Information for more details).

283  The CP pulse sequence acts as a quasi-monochromatic filter centered at $1/\tau$, so that a single
284  component of $b(t)$ can be sensed in each experimental realization. As a consequence, $P(n\tau, b)$
285  in Fig. 4(a) shows collapses occurring at $\tau \sim 1/2\nu_i$. Notice that the collapse corresponding to
286  the frequency component $\nu_{+1}$ ($\tau \simeq 3.448$ μs) cannot be resolved from noise since the first har-
287  monic of the filter function roughly coincides with the NSD peak ($\nu_{+1} \simeq \nu_L/3$) [Fig. 4(b)]. To
288  detect the three components of the target signal and filter out the NSD, we need an optimized
289  sequence. We thus apply the optimization algorithm detailed before to solve this experimental
290  sensing problem.

291  In order to confirm the theoretical prediction on how the optimized DD sequence can
292  outperform the standard control in terms of sensitivity, we performed measurements of the
293  sensitivity itself. Specifically we used three different CP sequences, each with time between
294  pulses $\tau = \frac{1}{2\nu_i}$, for $i = -1, 0, +1$. Having a previous knowledge of the NSD allows us to
295  predict the sensitivity of the the spin sensor using equations (3), (2), and (4), for any given
296  DD sequence, and for any target AC signal $b(t)$. In Fig. 5a we show the estimated values
297  for the inverse of the sensitivity as a function of the sensing time $T = n\tau$. Since $\tau = \frac{1}{2\nu_i}$ is
298  fixed for each of the CP sequences, the variation of $T$ corresponds to a variation of the number
299  of pulses $n$. Notice how for $\tau = \frac{1}{2\nu_{+1}}$, the inverse of the sensitivity rapidly goes to zero.
300  The estimated inverse sensitivity for the optimized sequence sign(SM)+SA is also shown in
301  Fig. 5a. The inverse sensitivity increases as a function of $T$, although we expect it to decrease
302  at longer times due to decoherence. In particular we know that for NV spin qubits the spin-
303  lattice relaxation time $T_1$ ultimately limits the sensing time $T$. However, even at shorter times
304  $T < T_1$ the sensitivity could be limited by other experimental factors, the most probable one
305  being $\pi$-pulse imperfections.

306  In the experiment, we measure $P(T, b)$ as a function of the field amplitude $b$ at a fixed
307  sensing time. An example of this kind of measurements is shown in Fig. 5b. From the analysis
308  of the oscillation of $P(n\tau, b)$, we can directly fit the values of $\chi$ and $\varphi/b$ (see Eqs. (1) and
309  (2)), and therefore we can obtain the values of $\eta$ using Eq. (4). The sensitivity measured
310  experimentally shows an excellent agreement with the expected simulated values (see Fig 5a).
311  See Supplementary Information for two additional test cases: one for a monochromatic target
312  signal such that the fifth harmonic of the NSD coincides with the frequency of the target signal;

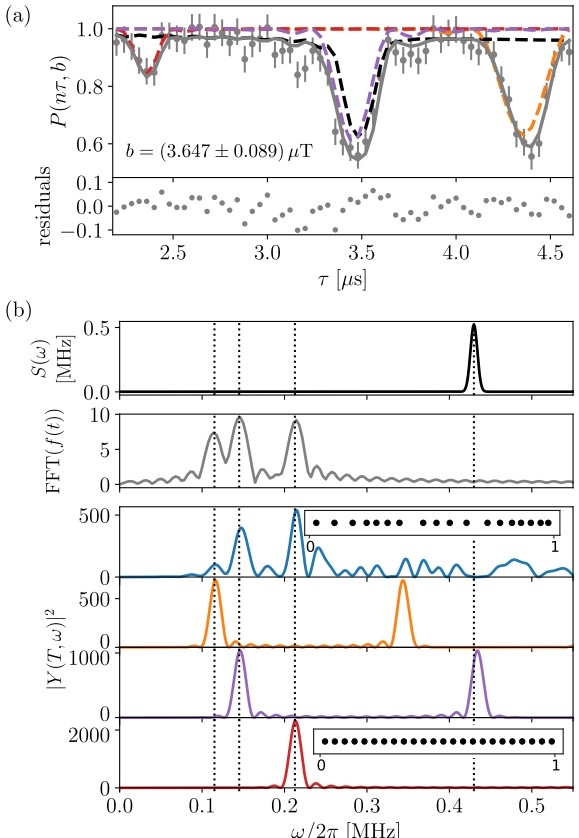

Figure 4: (a) Dynamics of the NV spin qubit under a DD sequence with $n = 16$ equidistant pulses (CP) for a trichromatic signal (see text). The NV spin coherence is mapped onto the probability of the NV spin to be in the state $|1\rangle$), $P(n\tau, b)$. Gray bullets: experimental data. Black dashed line: simulated spin coherence in the presence of noise, without any external target signals. Orange, red, and purple dashed lines: simulated spin coherence in the presence of monochromatic target fields with $\omega_1$, $\omega_2$, and $\omega_3$, respectively, with no noise. Gray solid line: simulated data combining all of the above using Eq. (1). Residuals between gray experimental data and gray solid line are shown in the bottom plot. (b) NSD given by the nuclear spin environment of the NV sensor (black line); fast Fourier transform (FFT) of the target signal $h(t)$ (gray line). Vertical dotted lines: frequency components of the target signal, and center of the NSD. Orange, purple, and red lines: filter function for a CP sequence with $T = \frac{n}{2\nu_i}$, for $i = -1, 0,$ and $+1$, respectively. Blue line: filter function of the optimized sequence. Inset: examples of time distribution of $\pi$ pulses.

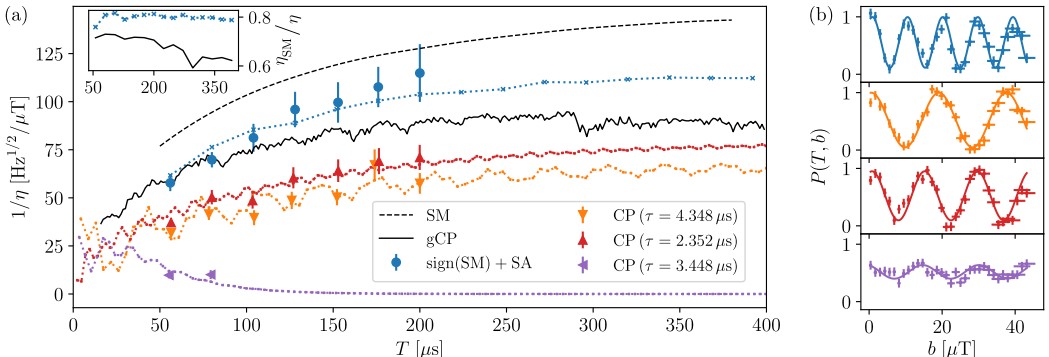

Figure 5: (a) Experimental values of the inverse sensitivity for the optimized sequences (blue circles; for $\Delta t = 160$ ns), and for the CP sequences (orange, red, and purple triangles). The predicted values of $1/\eta$ are represented by dotted lines. Black dashed line: theoretical upper bound of $1/\eta$, obtained from the solution of the spherical model in the continuum limit. Solid black line: predicted values of $1/\eta$ for the gCP sequence. Inset: Ratio $\eta_{SM}/\eta$ for the sign(SM)+SA and for the gCP sequences (blue and black, respectively). (b) $P(T, b)$ as a function of the amplitude of $b$, at $T \simeq 152 \, \mu s$. Same color code as in (a). Lines are a cosine fit (see text).

and one for a target signal with seven frequency components, all close to the NSD peak. These two cases confirm the results of the experiments shown in the main text.

## 5 Conclusion

We have shown that the problem of finding an optimal solution to quantum control a single spin system for quantum sensing can be solved by first finding the ground state of a solvable spherical model of classical spins, and then using this as a starting point for a simulated annealing algorithm. In this way, the optimization algorithm is able to find a control sequence that shows a significant improvement to the sensitivity with respect to standard control sequences. In addition, from the spherical model we found a theoretical bound on the sensitivity. Although the spherical model can be mapped to a control sequence that gives relatively good results, using the simulated annealing algorithm is necessary to improve even further the sensitivity, approaching **80−85%** of the bound. The fact that this result is consistent over the ensemble of cases studied numerically leads us to believe that an empirical bound for the sensitivity occurs at $\simeq 1.2\eta_{SM}$. Our experimental results confirm the theoretical predictions, hence validating our algorithm as an optimization protocol applicable to single spin sensors.

The proposed algorithm can solve the problem of finding the optimal DD sequence of a given signal $b(t)$ in a few *milliseconds* on a Raspberry Pi, which opens the door to miniaturization of the control electronics, using for example low-power processors. Fast optimization would also enable the implementation of adaptive protocols for sensing and spectroscopy.

## Acknowledgements

The authors would like to thank Marco Zennaro and Carlo Fonda for providing the Raspberry Pi microcomputer where the optimization algorithm was run. We also thank Francesco Poggiali for useful discussions, and Massimo Inguscio for constant support.

**Author contributions**   P.C., N.F., and S.H.-G. designed the experiment; S.H.-G. carried out the experiment and collected the data; G.F., F.B., and A.S. carried out the theoretical analysis and conceived the optimization protocol; S.H.-G. and F.B. analyzed the data. All authors have contributed equally to the writing of the manuscript.

## A   Definition of the sensitivity

In the main text, Eq. (4) introduced the sensitivity $\eta$ as the minimum detectable signal for unit time in our experimental platform. To justify this statement, here we sketch a brief derivation using both a direct approach, and a more formal one through the Fisher information.

First, let us define $\eta$ as the signal strength yielding a signal-to-noise ratio $\mathbf{SRN = 1}$ for a total experiment time of 1 s. Following Ref. [1], the SNR for $N$ independent experiments can be defined as

$$\mathrm{SNR} = \frac{\delta P(T, b)}{\sigma_N}, \tag{A.1}$$

where $\sigma_N$ encompasses all the sources of error, and $\delta P(T, b)$ is the spin population difference between the cases with and without target signal: $\delta P(T, b) = P(T, b) - P(T, 0)$. Now, the error can be shown to be of the form $\sigma_N \approx C^{-1}/\sqrt{N}$, with a dimensionless constant $C = O(1)$ depending on the experimental platform [1]. Also, using Eq. (1) of the main text, and assuming slope detection, one gets to

$$\delta P(T, b) \approx e^{-\chi(T)} \left| \sin\left(\varphi(T, b)\right) \frac{\partial \varphi(T, b)}{\partial b} b \right| = e^{-\chi(T)} |\varphi(T, b)|. \tag{A.2}$$

Thus, imposing $\mathbf{SNR \equiv 1}$ one finds

$$1 = e^{-\chi(T)} |\varphi(T, b)| \frac{1}{C\sqrt{N}} \tag{A.3}$$

and finally, using that one performs $N$ experiments in 1 s in total,

$$\eta = \frac{e^{\chi(T)}}{|\varphi(T)/b|} \sqrt{T}, \tag{A.4}$$

with $T$ being the time for a single experiment, and $C$ set to unity. This is exactly Eq. (4) of the main text.

As anticipated above, the sensitivity can be defined also through the Fisher information and the Cramér-Rao bound. Specifically, we define $\eta$ to be the minimum signal that can be distinguished from 0 in a total time of 1 s. Assuming that our estimator of the magnetic field $b$ is unbiased, from the Cramér-Rao bound it must be

$$\Delta b \geq \frac{1}{\sqrt{F_N}}, \tag{A.5}$$

where $F_N$ is the Fisher information associated with $N$ measurements of the magnetic field strength $b$ from an estimator $x$ [41, 47]:

$$F_N = \sum_x \frac{1}{p_N(x|b)} \left( \frac{\partial p_N(x|b)}{\partial b} \right)^2. \tag{A.6}$$

In our case, since we detect the $|\pm\rangle$ states in a Ramsey interferometry experiment, it holds $p(\pm|b) = \mathrm{Tr}(\rho |\pm\rangle\langle\pm|)$ with

$$\rho = \begin{pmatrix} 1/2 & e^{-\chi(T)-i\varphi(T,b)/2}/2 \\ e^{-\chi(T)+i\varphi(T,b)/2}/2 & 1/2 \end{pmatrix}, \tag{A.7}$$

and thus

$$F = \frac{8\varphi^2(T,b)}{b^2} \frac{e^{-2\chi(T)} \sin^2 \varphi(T,b)}{1 - e^{-2\chi(T)} \cos^2 \varphi(T,b)}. \tag{A.8}$$

Assuming slope detection, and for $N$ repeated measurements,

$$F_N = N \frac{8\varphi^2(T,b)e^{-2\chi(T)}}{b^2}, \tag{A.9}$$

since the Fisher information is additive for independent trials. At this point, recalling that the $N$ experiments have to be done in a total time of 1 s, and using the Cramér-Rao bound Eq. (A.5), one easily gets to Eq. (A.4), that is Eq. (4) of the main text.

# B   Details on the experimental platform

The ground state of an NV center is a spin triplet $S = 1$, naturally suited for sensing magnetic fields via Zeeman effect. The NV electronic spin presents extremely long coherence times, of the order of milliseconds at room temperature [13], due to the protective environment provided by the diamond itself. The $S = 1$ electronic spin can be initialized into the $m_S = 0$ state by addressing the NV center with green light (532 nm). This is due to an excitation–decay process involving radiative (637 nm) and non-radiative decay routes, occurring with a probability that depends on the spin projection $m_S$. This same mechanism implies that the red photoluminescence intensity of the $m_S = 0$ state is higher than the one of $m_S = \pm 1$, hence enabling to optically readout the state of the system. In addition, the internal structure of the NV center removes the degeneracy between the $m_S = \pm 1$ states and the $m_S = 0$ state, imposing a zero-field-splitting of $D_g \simeq 2.87$ GHz. An external bias field, aligned with the spin quantization axis, removes the degeneracy between the $m_S = \pm 1$ states, allowing to individually address the $m_S = 0 \leftrightarrow m_S = +1$ transition using on-resonance microwave radiation. By using microwave pulses with a appropriate duration, amplitude and phase, it is possible to apply any kind of gate to the single two level system. Therefore, the two level system formed by the $m_S = 0$ ($|0\rangle$) and $m_S = +1$ ($|1\rangle$) states fulfills the requirements to be used as a qubit based magnetometer.

## B.1   Characterization of the amplitude of the target signal

The target signal is delivered via a signal radio-frequency (RF) generator connected to the same wire, placed close to the diamond, that delivers the MW control field. We can control the amplitude of the target field by changing the output amplitude of the RF generator. However, the absolute value of the amplitude of the target field $b$ has to be characterized in order to take into account the attenuation of the circuit, the emission efficacy of the wire (which depends on the RF frequency) and the distance between the wire and the NV defect. To achieve such characterization, as explained in the main text, we measure the spin dynamics for a CP sequence as a function of the sequence interpulse time, and we compare with the simulation to minimize the residuals using $b$ as the only free parameter. By performing this measurements for different values of the RF generator output amplitude $a_{RF}$, we can extract a relation between $a_{RF}$ (in [Vpp]) and the amplitude of the target magnetic field $b$ (in [T]).

# C   Additional test cases

In order to reinforce our results, we repeated the analysis presented in the main text for two different target signals. A monochromatic target signal that coincides with one of the NSD

harmonics, and a 7-chromatic target signal that accentuates the difference between the generalized CP and the optimal solution.

## C.1   Second test case: Monochromatic target signal

If we want to detect a monochromatic target signal $b(t)$, in most cases a Carr-Purcell CP sequence of equidistant pulses is the best way to increase the sensor's response to that target signal and filter out the noise. This is due to the quasi-monochromatic filter function associated with a CP sequence. Assuming that $\tau$ is the time between pulses, the filter function shows a peak centered at $\omega/2\pi = \frac{1}{2\tau}$. However, the filter function is not exactly monochromatic, it shows harmonics at $\omega/2\pi = \frac{1}{2(2\ell+1)\tau}$, with $\ell \in \{1, 2, ...\}$. Therefore, if the frequency associated with $b(t)$ is close to $\omega_{\mathrm{L}}/(2\ell + 1)$, then a CP sequence will amplify the effect of both, the target signal and the noise, leading to not-optimal sensitivities.

Here we used the optimization algorithm described in the main text in order to obtain optimal sequences for this problem. In particular, we explored the case of a monochromatic signal with frequency $\nu_{\mathrm{mono}} = 39.29$ kHz, which is close enough to $\nu_{\mathrm{L}}/11$ so that the 5-th harmonic of the CP sequence coincides with the noise components. We used the same NSD $S(\omega)$ as in the three-chromatic case. The experimental values of $1/\eta$ are obtained from the measurement of $P(T, b)$ as a function of $b$. The results of $P(T, b)$ for one value of the sensing time $T$ are shown in Fig. 6(a). The predicted values of the inverse sensitivity, together with their experimental values are shown in Fig. 6(b). Similarly to the case detailed in the main text, the optimal sequences improve the sensitivity of the quantum sensor, resulting in some cases to an inverse sensitivity that is close to a twice the one from the CP sequence. In the monochromatic case explored here, the sensitivity gets worse when increasing the sensing time beyond $100\ \mu s$. Instead the optimal solutions are able to improve the sensitivity even for times $T > 300\ \mu s$. For $T \simeq 100\ \mu s$, and longer sensing times, the optimized sequences achieve higher values of $1/\eta$ than the maximum value achieved by a CP sequence.

## C.2   Third test case: 7-chromatic target signal

We have explored the case of a target signal with 7 frequency components, as specified in Fig. 7 (a-b). As in the main text, we used the optimization algorithm either to find the approximated spherical solution, or the solution using simulated annealing (SA) in order to minimize the sensitivity. The predicted values of the inverse sensitivity, together with their experimental values are shown in Fig. 7(c). Similarly to the previous test cases, the optimal sequences improve the sensitivity of our quantum sensor. In this case, the sensitivity obtained with the optimal solutions almost $1/2$, and $1/3$ with respect to the generalized CP (gCP) sequence for $T = 80\ \mu s$, and $T = 160\ \mu s$, respectively.

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
