# Peer review of "Optimal control of a quantum sensor: A fast algorithm based on an analytic solution"

_SciPost Physics_

## Round 1 · Referee Report · Anonymous (Referee 1) · 2024-2-29

Strengths

1- relevant physics issue addressed 2- original mapping to a known model 3- helpful discussion of approximations 4- experimental verification

Weaknesses

1- clarity of some formulations should be improved 2- final measurement of field strength not or insufficiently presented 3- not all sentences are in idiomatic English

Report

The authors study an interesting question, namely how one can
optimize the measurement sensitivity in presence of noise by
bang-bang control. The line or argument is theoretically appealing
since it eventually resorts to a frustrated 1D Ising model
and a mean field treatment known from spin glass theory.
Another great asset consists in the experimental verification
which the authors carried out as well. Thus, I am
voting for publication, once the points below will have been
addressed.

Requested changes

1) It must be emphasized early on that h(t) must be known beforehand. Only the amplitude can be detected. Please discuss whether this is a realistic situation occurring in practice! Me, being a theorist, am a bit skeptical whether the by far more relevant goal would not be to detect b(t) including its time dependence.

Similarly, please state clearly, that the power spectrum S(\omega) of the noise needs to be known a priori.

2) Emphasize early on that you are dealing with dephasing only, omitting other decoherence processes. Justify that "dephasing only" is still a relevant issue.

3) What is meant by the "overlap" between noise and target field between Eq. (3) and (4)? Please be a bit more precise.

4) The quantity \eta is called "sensitivity", but later minimized. From the sense of the word, however, the sensitivity should be large - so I would rather call 1/\eta sensitivity.

5) In Eq. (11) the inverse of a delta-function is used in the continuum. Please comment on what this really means and how one can define it mathematically.

6) After Eq. (15) please state that you take lambda to be constant. The notation lambda(t)=lambda is not unambiguous.

7) Fig. 2 should be rendered much larger, including the fonts, to reach a decent readability.

8) An important point needs to be elucidated in the experimental part: How can the "true" value of b read off and with which error? This must be elucidated, at best by a figure.

9) Fig. 6, panel b) the tick labels on the y-axis should start at 0.

10) Why does the rightmost panel of Fig. 7 not contain data for Sph.?

11) Finally, a careful reading by an English native speaker can improve the manuscript by leading to more idiomatic sentences at several occasions. "A minima" should read "A minimum", on page 9.

---

## Round 1 · Referee Report · Anonymous (Referee 2) · 2024-3-25

Strengths

1 - The connection between sensitivity and Hamiltonian minimisation is ingenious and potentially seminal
2 - Detailed implementation of the optimisation routines
3 - Experimental confirmation already implemented

Weaknesses

1 - The fact that the knowldge of the profile function h(t) must be assumed somewhat limits the practical impact

Report

As highlighted in strength and weaknesses above, this manuscript reports on a well thought-out and worthwhile study, with a clear transformative elements. Up to response to the comments set out below, I amd therefore glad to recommend its publication in current form.

Requested changes

1 - The overall clarity of the manuscript would be much increased if the abstract already specified the usage of the word "optimal", with a sentence to the effect of "in the sense that it optimises the sensitivity, i.e., the smallest detectable signal". It would also be beneficial to explicitly state in the introduction that the minimisation of the sensitivity is analogous to the classical Hamiltonian minimisation (upon time discretisation).

2 - In deriving (A.4), why is it that C can be set to 1?

3 - In Eqs.(2,4), I take it Phi(T)=Phi(T,b)/b ? (Since Phi(T,b) is proportional to b?) This shouldbe clarified.

4 - Eq.(4): absolutely explain the opernational significance of the sensitivity

---

## Editorial Decision

resubmitted